# An Overview of Thermal Exposure on Microstructural Degradation and Mechanical Properties in Ni-Based Single Crystal Superalloys

**DOI:** 10.3390/ma16051787

**Published:** 2023-02-22

**Authors:** Jian Zhang, Fan Lu, Longfei Li

**Affiliations:** 1Science and Technology on Advanced High Temperature Structural Materials Laboratory, AECC Beijing Institute of Aeronautical Materials, Beijing 100095, China; 2State Key Laboratory for Advanced Metals and Materials, University of Science and Technology Beijing, Beijing 100083, China; 3Beijing Key Laboratory of Special Melting and Reparation of High-End Metal Materials, University of Science and Technology Beijing, Beijing 100083, China

**Keywords:** Ni-based single crystal superalloys, thermal exposure, microstructural evolution, TCP phases, mechanical property

## Abstract

Microstructural stability at elevated temperatures is one of the main concerns for the service reliability of aero-engine turbine blades. Thermal exposure, as an important approach to examine the microstructural degradation, has been widely studied in Ni-based single crystal (SX) superalloys for decades. This paper presents a review on the microstructural degradation induced by high-temperature thermal exposure and the associated damage in mechanical properties in some typical Ni-based SX superalloys. The main factors affecting the microstructural evolution during thermal exposure and the influencing factors in the degradation of mechanical properties are also summarized. Insights into the quantitative estimation of the thermal exposure-affected microstructural evolution and the mechanical properties will be beneficial for the understanding and improvement of reliable service in Ni-based SX superalloys.

## 1. Introduction

Ni-based single crystal (SX) superalloys are the materials of choice for the manufacturing of turbine blades in aero-engine and power-generation applications due to their unique high temperature performance [1]. During service, the turbine blades are subjected to extreme conditions such as high temperature, changeable mechanical stresses and environmental corrosion. Hence, the initial microstructures undergo an inevitable degradation process at high temperature, which contributes much to the decline in component performance and even the premature failure of the gas turbine blades [2,3]. The strength of a given Ni-based SX superalloy mainly refers to its mechanical properties as well as microstructural stability [4]. As the direct approach to estimate microstructural stability, knowledge of thermal exposure on microstructural evolution and the related mechanical properties is extremely essential to further optimize the alloying design and achieve superior performance during service of Ni-based SX superalloys.

The outstanding properties of Ni-based SX superalloys can be attributed to the microstructures combining the ordered (L1_2_) intermetallic γ’-Ni_3_Al precipitates, coherently embedded in disordered (fcc) γ-Al matrix [5,6]. The γ’ phases with a high volume fraction provide high rigidity and low dislocation tolerance, limiting the dislocation movements in the γ channels, which contribute to the high temperature property by the precipitate strengthening effect [7]. Thus, the volume fraction, size, morphology and distribution of the γ’ precipitates become the primary concerns in the microstructural evolution when serving under harsh conditions [8,9,10].

To meet the requirements in high-temperature capabilities, modern Ni-based SX superalloys are always alloyed with amounts of refractory elements, such as W, Mo and, most importantly, Re [5,11,12]. The introduction of these elements, however, can also promote the rapid formation of refractory-rich topologically-close-packed (TCP) precipitates belonging to brittle inclusions at elevated temperatures, which may significantly damage the endurance life of Ni-based SX superalloys [13,14,15,16]. Thus, another great concern in evaluating microstructural degradation is the precipitate of TCP phases [17,18,19].

Obviously, the microstructure degradation induced by thermal exposure, including the changes in the volume fraction, size, morphology and distribution of the γ’ precipitates as well as the formation of TCP phases, is critical to the mechanical properties [20,21]. Since different alloys present microstructure degradation at different extents when subjected to different conditions, or even the same conditions, it is important to summarize the microstructural degradation in different aspects for a better understanding of the microstructural stability of modern Ni-based SX superalloys. Therefore, the alloying design can be further optimized when comprehensively considering the microstructural stability of the characterized Ni-based SX superalloys. The purpose of this paper is to review the high temperature characteristics and evolution of γ/γ’ phases and the formation of TCP phases as well as the influencing factors and their effect on related mechanical properties based on published experimental results of several modern Ni-based SX superalloys.

## 2. Microstructural Stability during Thermal Exposure

### 2.1. γ/γ’ Microstructure Evolution

The most remarkable microstructural evolution in Ni-based SX superalloys exposed to high temperature is the growth of the γ’ phases, or so-called coarsening [22,23]. In the two-phase mixed system, a large amount of γ/γ’ interfacial area is the direct consequence of the polydisperse nature. During thermal exposure, the total energy of the system should be decreased to reach the equilibrium state. Thus, by decreasing the amount of interfacial area, the system tends to have a thermodynamic preference, resulting in the growth of the precipitate size [24]. Figure 1 shows the typical γ/γ′ microstructure of a commercial Ni-based SX superalloy CMSX-4 after heat treatments (referring to initial state) and after related thermal exposure at 950 °C for different durations [25]. In the initial state, γ′ precipitates exhibit square morphology on each {001} crystallographic plane, illustrating the cuboidal morphology before thermal exposure. During thermal exposure, the γ′ precipitates increase their average sizes accompanied by some elongated precipitates, although most of the precipitates still maintain their straight edges and sharp corners. This represents the most common microstructural evolution of γ′ precipitates when subjected to high-temperature thermal exposure, indicating the shrink of small particles and the growth of large particles, followed by the well-known Ostwald ripening theory [22,26].

More serious γ′ evolution may occur from an initial cuboidal shape into a plate-like morphology aligned along the <100> direction during a stress-free thermal exposure process, or so-called spontaneous rafting. Figure 2a shows the morphology of the spontaneous rafting in a DD11 alloy when subjected to 1070 °C and 300 h [27]. It is confirmed that this phenomenon is closely associated with a higher γ′ volume fraction, which leads to the easier interconnection of the growing γ′ phases in the coarsening process [28]. Furthermore, the addition of a large amount of refractory elements into Ni-based SX superalloys causes considerable negative lattice misfit, which acts as the driving force for the directional diffusion flow and the formation of γ′ rafts aligned along the <100>, despite the absence of applied stress [29]. Meanwhile, the elevate temperature also contributes to the elemental diffusion process and further promotes the faster developed γ′ rafts [30,31]. For the modern Ni-based SX superalloys, designed with a considerable addition of refractory elements, the spontaneous rafting is also a representative feature in microstructural instability in many published works [32,33,34].

It is well-known that in superalloys with extremely high γ′ volume fraction, the γ′ phases tend to be seriously interconnected and can even become the topological matrix phase instead of the γ matrix during creep, wherein this process can be called topological phase inversion [35]. Recent research also displayed a similar phenomenon in the interdendritic regions of specimens after thermal exposure, as shown in Figure 2b [36]. This strongly confirmed the effect of the higher γ′ volume fraction as responsible for the serious microstructural evolution, since the γ′ volume fraction in the interdendritic regions is over 65%, which is much higher than that in the dendrite core.

**Figure 2 materials-16-01787-f002:**
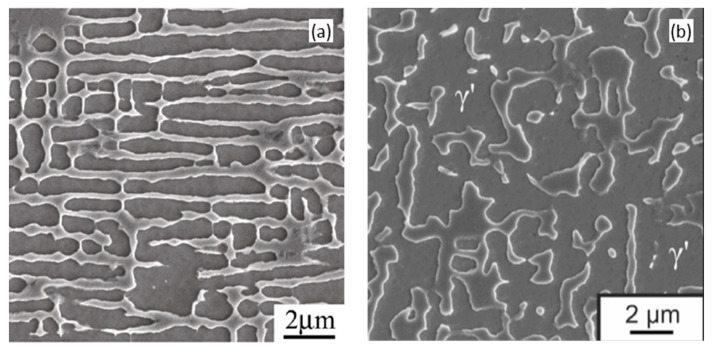
The formation of (**a**) spontaneous rafts in DD11 alloy after thermal exposure of 1070 °C/300 h [27] and (**b**) topological phase inversion in interdendritic areas in ERBO/1 alloy after thermal exposure of 1100 °C/250 h [36].

Figure 3 shows the quantitative microstructural parameters (including the γ′ volume fraction and γ′ and γ phases’ dimensions) of the DD11 alloy after thermal exposure at 1070 °C for different durations. It is suggested that all the microstructural parameters changed gradually until 500 h, after which they tend to approach a constant value under the specific temperature. Here, the dissolution of γ’ is a diffusion-controlled process, following a model from Johnson–Mehl–Avrami–Kolmogorov (JMAK), which predicts the real-time γ′ volume fraction as a function of the time [37,38]. Above all, the decreasing γ′ volume fraction and increasing γ and γ′ sizes constituted the important features of γ/γ′ microstructural degradation during the thermal exposure process.

### 2.2. γ’ Coarsening Mechanisms

To quantitatively describe the precipitate coarsening kinetics, a theory was first proposed by Lifshitz and Slyozof [23] and then by Wagner [26], which is widely known as the LSW theory. The remarkable feature of the developed theory is characterized by the time exponents in the kinetics of precipitate coarsening, which is essential to diffusion-controlled coarsening. Based on the Gibbs–Thomson equation, the LSW theory provided an analytical equation: $\langle r^{n}\rangle =Kt$, with the exponent of n = 3, where 〈r〉 is the average precipitates radius, K is the coarsening constant and t is the related time. Figure 4b gives the plot of $\langle r^{3}\rangle$ versus t of a Ni-Al binary alloy at different temperatures, indicating the accepted goodness-of-fit in the presumed linear behaviors. However, the LSW theory was developed based on binary systems and assumed a dilute system with small volume fraction of precipitates. Thus, some followed models were developed with the consideration of high precipitate volume fraction, such as Modified LSW (MLSW) [9], Davies–Nash–Stevens (LSEM) [39], Brailsford–Wynblatt (BW) [40], etc. More recently, a new model was proposed by Philippe and Voorhees (P-V), taking into account the multi-component effect as well as the precipitate volume fraction [41]. Nevertheless, the LSW theory is still the most well-accepted theory to describe the coarsening behavior, and a large number of studies have successfully expanded the LSW theory to multi-component superalloys.

Another coarsening model was established which considered that the diffusion through the interface controlled the coarsening process, called the trans-interface diffusion-controlled (TIDC) model. This model excludes the effect of volume fraction while taking the interfacial width into account and the exponent of n = 2 in the mentioned power-law function. Figure 4a outlines the relationship between $\langle r^{2}\rangle$ versus t, also suggesting good linear behavior, which adapted to the TIDC theory. Recently, the TIDC theory has also been successfully applied for multi-component superalloys [43,44,45].

The two coarsening controlled mechanisms, referring to matrix-diffusion controlled and interfacial-diffusion controlled, usually emerge simultaneously and compete in a single coarsening process [46]. By the analysis of the particle size distributions (PSDs) within different coarsening stages, the underlying coarsening-controlled mechanism can be determined. Figure 5 shows the experimentally measured PSDs as well as the predictions of PSDs for LSW and TIDC theory in a Ni-Al-Cr-Re alloy, firstly with lower temperature, which is similar to the stage before heat treatment of the experimental alloy, followed by a Re-containing alloy with higher temperature and longer aging time [44]. It is indicated that, in the heat treatment process, the coarsening is mainly controlled by matrix diffusion, since the PSDs are much closer to the prediction in LSW theory while, at a longer thermal exposure time, the coarsening gradually tends to be controlled by interfacial-diffusion. More recent research also confirmed the transition of the coarsening-controlled mechanism from matrix-diffusion to interfacial-diffusion in the long-term coarsening process of a single alloying system, where the decreasing interfacial width acts as the driving force for that transition [45].

### 2.3. Precipitation Behavior of TCP Phases

Excessive addition of refractory elements promotes the precipitate of the topologically close-packed (TCP) phases, such as σ-, μ-, P- or R-phase, during high-temperature thermal exposure in Ni-based SX superalloys [19]. These TCP phases can display a wide variety of different morphologies, as shown in Figure 6 [47,48]. In 2D characterization, σ phases and P phases mainly exist in sheet-like and needle-like morphology (Figure 6a), while in 3D characterization, the P phases present a basket-weave-like morphology, showing the possibility of growing from σ phase after thermal exposure (Figure 6c). The different morphology of TCP phase depends on the different crystallography structures. Furthermore, both of the σ and P phases show a consistent orientation relationship with the matrix, with different thermal exposure conditions.

The needle-like μ phases and Lath-like R phases are shown in Figure 6b,d. In 3D characterization, the basket-weave-like R phases consist of the intersected needle-like μ phase. The R phases serve as the thermodynamic equilibrium phase, which can precipitate from the matrix as well as μ phases, while it has no direct orientation relationship with the matrix.

The TCP phases basically contain a large number of γ-stabilizers, such as Re, W, Mo, Cr and Co, as shown in the APT characterization in Figure 7. Thus, the TCP phases are always surrounded by γ′ phases. While there are still some differences in chemistry among the TCP phases, usually, P and σ phases are very similar in high Re content, μ phases are high in Mo and/or W and low in Re content, and Co content in μ phases is much higher than that in σ phases [49]. Since the precipitates of TCP phases depend on a thermodynamic process, they can be qualitatively determined in CALPHAD calculations.

Figure 8a shows the differentiation within TCP phases’ formation between the dendritic and interdendritic region after thermal exposure of ERBO/1C alloy [51]. It indicated the pronounced formation of TCP phases in dendrite cores rather than the interdendrite region, which can be explained by the strong segregation of the refractory elements to dendrite cores after heat treatment of the alloy, especially Re. A threshold Re concentration is necessary for the formation of the TCP phase. The evolution of the TCP phase in Figure 8b also shows the sharp increase in the TCP area fraction in dendrite cores during thermal exposure. Thus, a compatible homogenous process is beneficial for decreasing the driving force for the TCP formation. However, the density of the TCP phases still decreases from the dendrite cores to the interdendritic regions due to the micro-segregation of refractory elements to the dendrite cores retained even after heat treatment [49,52].

### 2.4. Influencing Factors on Microstructural Stability

#### 2.4.1. Temperature and Time

Figure 9a shows the dependence of average size of the γ’ precipitates on the thermal exposure time of the CMSX-4 alloy. The average precipitate size increases with the thermal exposure time as well as temperature. Detailed analysis on the growing cuboidal γ’ precipitates showed that the coarsening kinetics follow a cube rate law, and higher temperature promotes the spontaneous rafts, as shown in Figure 9b. Figure 9c shows the temperature dependence of the experimental and calculated coarsening rate of CMSX-4 alloy, indicating the significantly increasing coarsening rate as a function of temperature. The active promotion of the coarsening rate by temperature can be attributed to the increasing diffusion rate at higher temperature [25].

The formation of TCP phases is also closely associated with the temperature and time. In the thermal exposure process of a Re-free alloy, the area fraction of the TCP phase increases with time, together with the decreasing incubation time and the increasing formation rate with the increasing temperature, as shown in Figure 10 [14]. This is mainly due to the higher diffusion rate of TCP formation elements suggested by higher temperature. However, the opposite conclusion has been obtained, indicating the restrained formation of TCP phases induced by increasing temperature, as shown in Figure 11a [54]. Here, although the TCP phases form much earlier with a shorter incubation time in higher temperature (1050 °C), the equilibrium content of TCP phases is obviously lower. Matuszewski et al. [52] presented CALPHAD calculations to predict the driving force for TCP precipitation at different temperatures, also indicating the decreasing driving force for promoting any type of TCP phase formation with the increasing temperature. This can be attributed to the following reasons: one is that the entropy term increases along with the increased temperature, which stabilizes the solid solution strength of the γ matrix, and another is that the γ volume fraction increases with the increased temperature, which restrains the supersaturation of refractory elements.

#### 2.4.2. Lattice Misfit and Interfacial Energy

It is widely known that the γ′ phases undergo the coarsening process when subjected to elevated temperature, with the reduction in total interfacial energy as the driving force. Previously, many researchers have been devoted to quantitatively estimating the interfacial energy of the alloys. The most accepted approach is to calculate it from the coarsening rate, with the function provided by LSW theory or the more recent P-V model [41,55]. Another available approach is to calculate it by the interfacial width and interfacial gradient with advanced characterization in Ardell’s method [56]. As the significant element in Ni-based SX superalloy, Re has received much attention in the calculation of the interfacial energy. Zhang et al. [57] summarized the interfacial energy of the alloys with Re addition and without Re, showing the obvious decrease in the interfacial energy induced by Re, as shown in Figure 12. This becomes the primary factor in the stabilized microstructural evolution by the addition of Re.

Another great concern is the lattice misfit. The alloy should be designed to a compatible composition leading to a reliable lattice misfit for suppressing γ′ coarsening. Although there is still no quantitative relationship between the lattice misfit and the interfacial energy, it can be concluded that interfacial energy may decrease with the decreasing lattice misfit (approach to zero), which has been clarified by a different alloying system [45,58]. Additionally, in Figure 12, there is a supportive tendency of the increasing interfacial energy along with the increasing value of the lattice misfit.

Liang et al. [59] conducted predictions of the interfacial energy induced by temperature using the CALPHAD method. In Figure 13a, the interfacial energy decreases by the increasing driving force, which restrains the driving force for γ′ coarsening, as shown in Figure 13b. Despite the decreasing interfacial energy by the increasing temperature, the elemental diffusion can be simultaneously promoted at higher temperature, which can, on the other hand, boost the coarsening behavior. Thus, the actual real-time coarsening behavior should consider both the interfacial energy and the elemental diffusion.

#### 2.4.3. Role of the Alloying Elements

The coarsening behavior as well as the TCP formation are all seriously affected by the alloying elements. Zhang et al. [57] summarized the coarsening rate as a function of Re content at various temperatures, as shown in Figure 14. The addition of Re can effectively reduce the γ′ coarsening rate at high temperatures, especially when the Re content increases from 0 to 4 wt.%.

The volume fraction of the TCP phase after thermal exposure of different alloys at 1000 °C for 1000 h has been summarized in Figure 15, which presents the effect of alloying elements on the TCP phase formation [60]. The addition of Mo and Re can sharply promote TCP precipitation, and Cr also plays a significant promoted effect. The TCP phase increases slightly by the increased W content. However, the addition of Co can effectively restrain the TCP formation. Interestingly, the commercial Ni-based SX superalloy CMSX-4 is always stable with respect to the TCP precipitate at elevated temperatures, although it has a considerable content of Re, W and Cr. This can be attributed to the lower content of Mo addition [12].

## 3. Effect of Thermal Exposure on Mechanical Properties

### 3.1. High-Temperature Tensile Strength and Room-Temperature Hardness

An examination of the related mechanical properties of the unexposed and exposed samples can be a valuable approach to determine the extent of the microstructural degradation. Figure 16a provides the typical strain–stress curves of CMSX-4 alloy after thermal exposure for different times [61]. The yield strength decreases along with the increased time. However, both UTS and the total elongation first increase firstly and then continuously decrease with the increasing time, as shown in Figure 16b. Another concern is the Vickers hardness. It can be seen that a continuous decrease in the Vickers hardness occurs along with the increased thermal exposure time in CMSX-4 alloy, indicating that all studied samples were overaged with respect to expected mechanical property, as shown in Figure 17 [61].

### 3.2. Creep Property

Since the evolution in volume fraction, size, morphology and distribution of γ′ phases as well as the TCP formation induced by thermal exposure mainly contribute to the degradation of creep resistance, another great estimation of the microstructural degradation is the related creep property. Figure 18 shows the creep lifetime of DD6 alloy at 1070 °C/140 MPa after thermal exposure at 980 °C or 1070 °C for different durations [34]. Note that no TCP phases formed in each specimen in this research. The microstructures with the longest thermal exposure time of 1000 h at different temperatures are shown in the corresponding figure, indicating the obvious rafting structure at 1070 °C/1000 h instead of the nearly cuboidal morphology at 980 °C/1000 h. In Figure 18a, obvious decreasing creep life occurs with the extended thermal exposure time until 400 h, which is then followed by the slight increase in creep life with the prolonged thermal exposure time. This is mainly due to the thermal exposure time (over 400 h) acting as the appropriate heat treatment process for the more proper growth of the γ′ phase to obtain the over-estimated property. However, after thermal exposure at 1070 °C for different durations, the related creep life exhibits a continuous decrease due to the serious degradation of γ/γ′ phases.

Cheng et al. [62] conducted a series of creep tests on the exposed specimens with TCP phases of CMSX-4 alloy, as shown in Figure 19. Here, more TCP phases formed at 1050 °C/2000 h than that at 950 °C/2000 h. This indicates the decreasing creep properties with increasing thermal exposure temperature or time. Obviously, when conducting thermal exposure at higher temperatures, the creep life exhibits wider separation between the thermal exposure time at 1000 h and 2000 h, which is mainly attributed to the significant increase in the TCP formation at 1050 °C/2000 h.

### 3.3. Influencing Factors on the Degradation of Mechanical Properties

Since the γ′ and γ phases play the leading role in the strengthening effect, which refers to the precipitate strengthening effect and solid solution effect, the γ/γ′ degradation should take the primary responsibility for the degradation of mechanical properties. From the microstructural aspect, the decreasing γ′ volume fraction and coarsening γ′ average size contribute to the decrease in Vickers hardness and tensile property [61]. Another deleterious factor is the increasing γ channel width during thermal exposure, which provides a decrease in Orowan resistance and leads to the increased dislocation slipping rate in the matrix [27].

The TCP phases are brittle inclusions which are composed of various refractory elements and have higher hardness than γ and γ′ phases. Hence, TCP phases are always seen to be greatly harmful for mechanical property. This is mainly due to two aspects: (1) TCP phases deplete the solid solution strengtheners (Re, W, Mo, Cr and Co) from the γ matrix, leading to the impaired solid solution strengthening effect of the alloy. (2) The loss of coherency at the TCP/γ′ phases becomes the initiating site of the micro-pores or even cracks. However, some controversies still remain concerning the second point. In Figure 20a, after creep within the exposed samples, the TCP phase showed good coherency with the surrounding γ′ phase, remaining free of crack but with only slight twist. In Figure 20b, although some micro-pores or crack initiation were found near the γ′ phase, they still did not propagate into macro-cracks [62].

Here, when the TCP phases remain in a low volume fraction, they may have no obvious effect on the failure in the creep tests except for depleting the solid solution elements. Sun et al. [63] also found that the deformation pores (D-pores) with small sizes have almost no clear relationship with the TCP phases. Even near the creep fracture, most fatal cracks are produced by initial pores in the interdendritic regions, rather than the TCP phases, as shown in Figure 21.

However, Zhang et al. [64] demonstrated the microcracks generating and propagating near the TCP phases with a direct angle of approximately 70° between the growth direction of the TCP phase and the microcrack, as shown in Figure 22a. This angle refers to the angle of the slip plane between (111) and (−1–11) as well as the angle of the slip direction between (−1–12) and (112). It is suggested that TCP phases promote the crack initiation at elevated creep temperature. Moreover, in the real tenon of a SX turbine blade, presented in Figure 22b, the macrocrack was propagated with a zigzag morphology, as predicted. The TCP phases, as the obstacles of the dislocation movements leading to the local pile-up of dislocations, also contribute to the crack initiation and propagation near the TCP phases. Although it is widely accepted that the effect of TCP phases on the deteriorated creep property mainly depends on the depleted solid solution elements, the promotion of crack initiation and propagation still cannot be excluded.

Ni-based SX superalloys are designed to exclude the effect of a weak grain boundary, and recent research has mainly focused on the microstructural evolution on γ/γ′ phases and TCP formation. However, it has been increasingly acknowledged that the γ/γ′ phases exhibit a different evolution tendency when suffering thermal exposure; thus, the microstructural evolution in the interdendrite region cannot be ignored [30,65]. Furthermore, the inevitable carbides in the interdendrite region may change and affect the mechanical properties. Huang et al. [32] found that the carbides can affect TCP formation where TCP phases precipitate preferentially within the vicinity of MC carbides. An et al. [66] also found changes in the type of the carbides during creep. The last concern is that the micro-pores formed after heat treatment exhibit growth during the following creep tests [67]. These factors can pose an even more serious effect on the creep property, to which significant attention should be paid in the study of the thermal exposure or creep of the superalloys.

Additionally, the formation of a rafting structure may also play an enhanced role in the mechanical properties. It has been recognized that the plate-like rafting structure can provide a longer distance for dislocations to climb, so as to impede the dislocation movements and enhanced creep property. During creep at high temperatures, the formation of a N-type rafting structure has always appeared at the minimum creep rate after the decreasing creep rate stage [68]. Although M.V. Nathal et al. [69] found that the pre-rafted structure would cause damage to the creep property, U. Tetzlaff et al. [70] conversely found that when achieving the P-type pre-rafted structure in compression, the iso-thermal fatigue strength and tensile creep property would be further enhanced. The enhanced ability depended on the applied stress, which determines the time available for the formation of the rafting structure introduced by the prior compressive creep strain.

Ni-based SX superalloys are still the materials of choice for the components serving at elevated temperatures and under load, where several modern Ni-based SX superalloys are usually considered, such as CMSX-4, René N5, DD6, et al. These alloys always contain a certain content of Re to improve the mechanical properties at high temperature. Thus far, the manufacturing technology of the SX superalloys has gradually matured and advanced, and many researchers have worked on these types of alloys and achieved lots of data, ensuring the safety of these alloys in real service. It can be deduced that the Ni-based SX superalloys will still be the primary choice in aero-engine blades, although the alloying design should be further optimized.

In summation, the γ/γ′ microstructural degradation and the formation of TCP phases are the primary aspects in influencing the microstructural stability of the Ni-based superalloys. Since the γ/γ′ microstructural degradation is driven by the interfacial energy, lower interfacial energy should be considered in optimizing the alloying design or, more directly, the lower lattice misfit approaching to zero. Another concern is the alloying additions, where a lower content of refractory elements induces lower driving force for the formation of TCP phases. Additionally, a lower content of refractory elements can also promote lower lattice misfit of the alloy. Generally, limiting the content of refractory elements is necessary to balance the microstructural stability and the mechanical properties of Ni-based SX superalloys.

## 4. Conclusions

In order to estimate the deterioration of microstructure in service, stress-free thermal exposure tests have become the most widely used approach in Ni-based single crystal (SX) superalloys. During high-temperature thermal exposure, γ/γ′ phases exhibit obvious degradation, displaying, as the γ′ volume fraction decreases, increasing γ′ sizes and a broadening γ matrix. A higher γ′ volume fraction can promote serious microstructural evolution, such as directional rafting and topological inversion. The growth in γ′ sizes driven by interfacial energy follows the traditional LSW theory as well as the newly developed TIDC theory, where the coarsening process is controlled by matrix-diffusion, at first followed by interfacial-diffusion at longer times. Due to the addition of refractory elements, the TCP phases may precipitate basically from dendrite cores in a variety of types, such as σ-, μ-, P- or R-phase. They usually contain a large number of γ-stabilizers such as Re, W, Mo, Cr and Co, and they can be distinguished by different morphologies and structures. The temperature, time, interfacial energy (and the associated lattice misfit) and alloying elements can pose different effects on the microstructural degradation during thermal exposure.

Microstructural degradation can lead to damage of the related mechanical properties, including tensile property, Vickers hardness and creep property. This is mainly due to the deteriorated precipitate strengthening and solid solution strengthening effect induced by the decreasing γ′ volume fraction, coarsening γ′ average size as well as the broadening γ channel. The formation of TCP phases can also result in a decrease in mechanical property along with the depleted solid solution elements, while the promotion of crack initiation and propagation still cannot be ignored. Further, the γ/γ′ microstructural evolution in the interdendrite region, the evolution of carbides and the formation and growth of micropores should also be carefully considered in the investigations of thermal exposure.

On this basis, in order to achieve better microstructural stability and the related mechanical properties of Ni-based SX superalloys, lower interfacial energy (as well as the lattice misfit approaching to zero) and a lower content of refractory elements should be considered for optimizing the alloying design.

## Figures and Tables

**Figure 1 materials-16-01787-f001:**
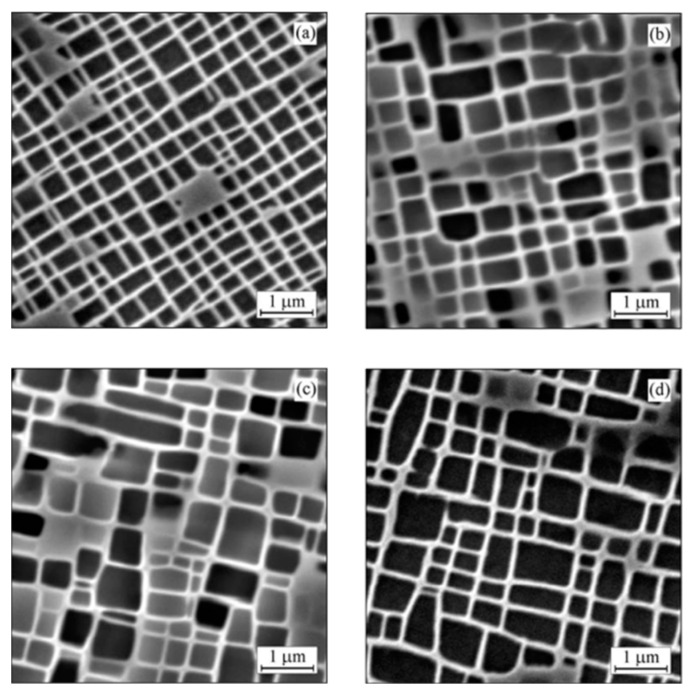
γ/γ’ microstructure evolution of CMSX-4 superalloy at (**a**) initial state; (**b**) 950 °C/500 h; (**c**) 950 °C/1000 h; (**d**) 950 °C/2000 h [25].

**Figure 3 materials-16-01787-f003:**
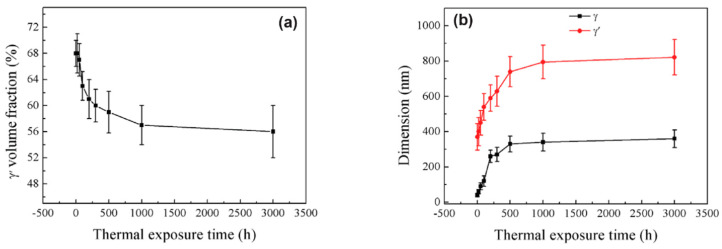
The quantitative microstructural parameters of DD11 alloy after thermal exposure at 1070 °C for different durations. (**a**) γ’ volume fraction; (**b**) γ’ size and γ channel width [27].

**Figure 4 materials-16-01787-f004:**
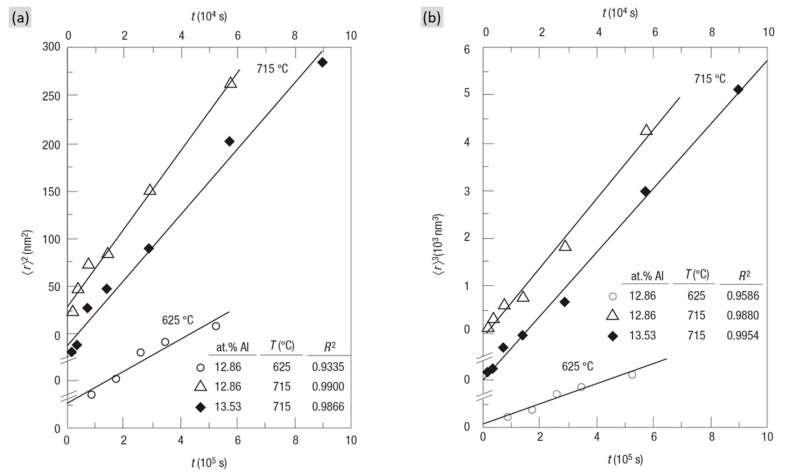
The plots showing the average γ’ precipitate size raised in (**a**) square rate law and (**b**) cubic rate law versus thermal exposure time in a Ni-Al binary alloy [42].

**Figure 5 materials-16-01787-f005:**
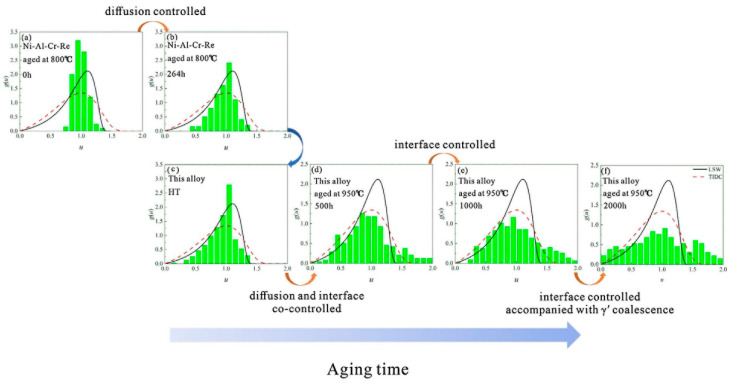
PSDs for Ni-Al-Cr-Re aged for (**a**) 0 h and (**b**) 264 h at 800 °C and PSDs for investigated alloy for (**c**) HT state and aged for (**d**) 500 h, (**e**) 1000 h and (**f**) 2000 h at 950 °C [44]. (u represent the normalized γ’ size).

**Figure 6 materials-16-01787-f006:**
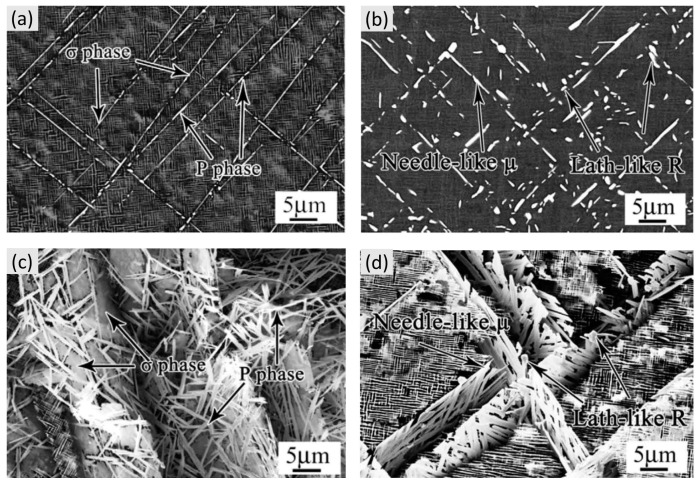
PSDs for Ni-A Typical TCP phases in Ni-based SX superalloys. (**a**) σ phase and P phase (2D-characterization); (**b**) μ phase and R phase (2D-characterization); (**c**) σ phase and P phase (3D-characterization); (**d**) μ phase and R phase (3D-characterization) [47,48].

**Figure 7 materials-16-01787-f007:**
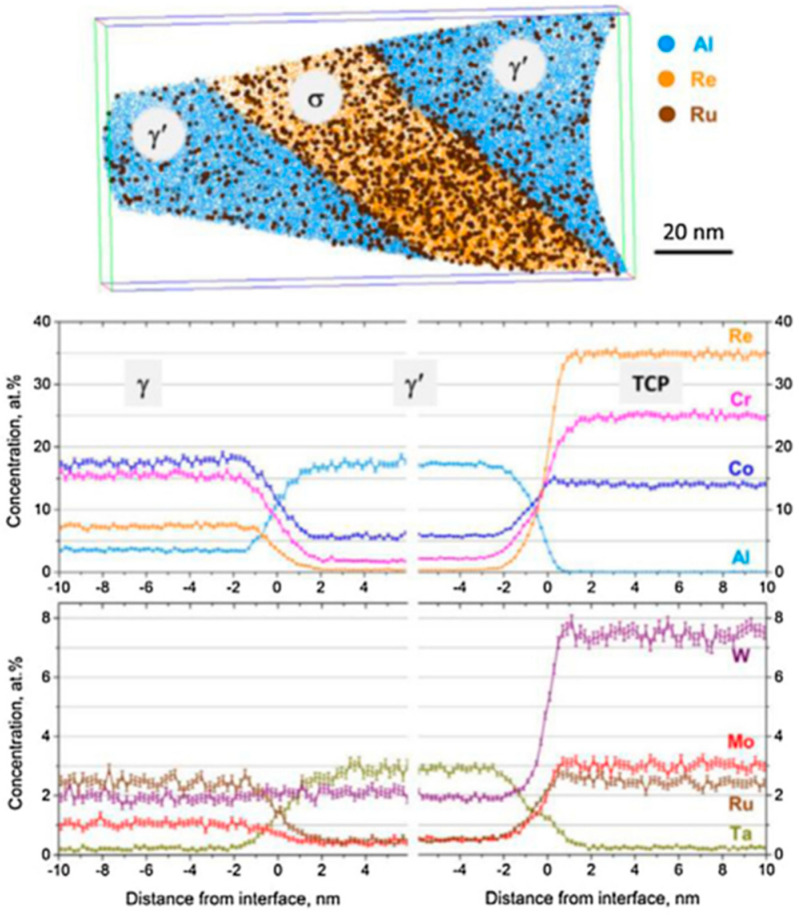
An APT elemental map of a plate-like σ precipitate surrounded by a γ’ envelope (only Al, Re and Ru atoms are shown) and corresponding concentration profiles across γ/γ’ and γ’/γ interfaces for the annealed Astra1-21 alloy. Accumulation of Cr, Mo, W and Re in the σ phase is visible. Ru concentrations in the σ phase and γ matrix are nearly identical. No elemental segregation at phase boundaries can be observed [50].

**Figure 8 materials-16-01787-f008:**
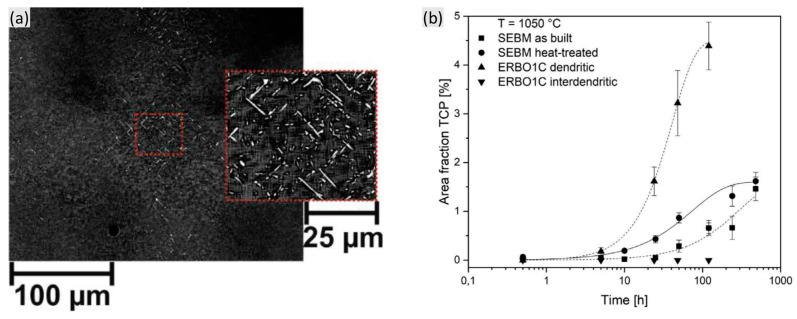
(**a**) Microstructure of the ERBO/1C alloy after thermal exposure at 1050 °C for 120 h and (**b**) area fraction of formed TCPs in dendritic core and interdendritic region [51].

**Figure 9 materials-16-01787-f009:**
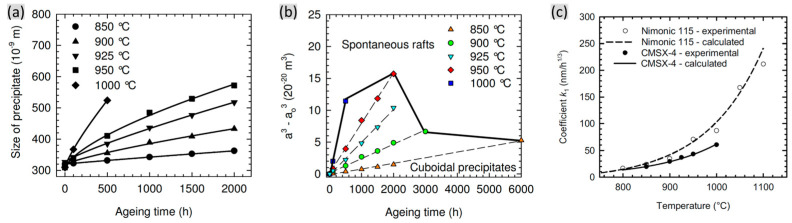
Dependence of the (**a**) average γ’ size and (**b**) normalized γ’ size as a function of thermal exposure time of CMSX-4 alloy after different durations at different temperature [53]. (**c**) Dependence of the coarsening rate as a function of temperature of two alloys [25].

**Figure 10 materials-16-01787-f010:**
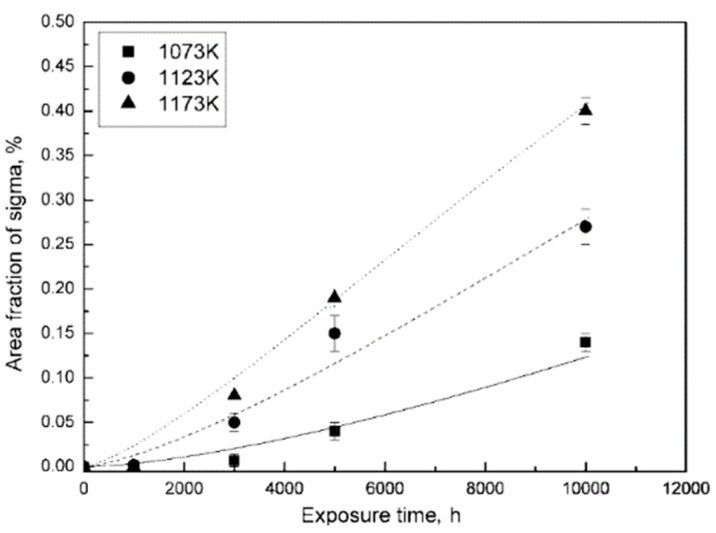
Variations in TCP phase fraction as a function of thermal exposure time in a Re-free Ni-based superalloy [14].

**Figure 11 materials-16-01787-f011:**
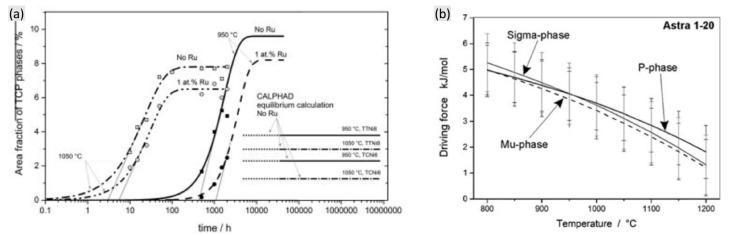
(**a**) Area fraction of TCP phases as a function of time for the alloys Astra 1–20 and Astra 1–21 at 950 °C and 1050 °C [54] and (**b**) the driving force for the precipitation of distinct phases within the alloy Astra 1–20 containing 2 at% Re with varying temperatures [52].

**Figure 12 materials-16-01787-f012:**
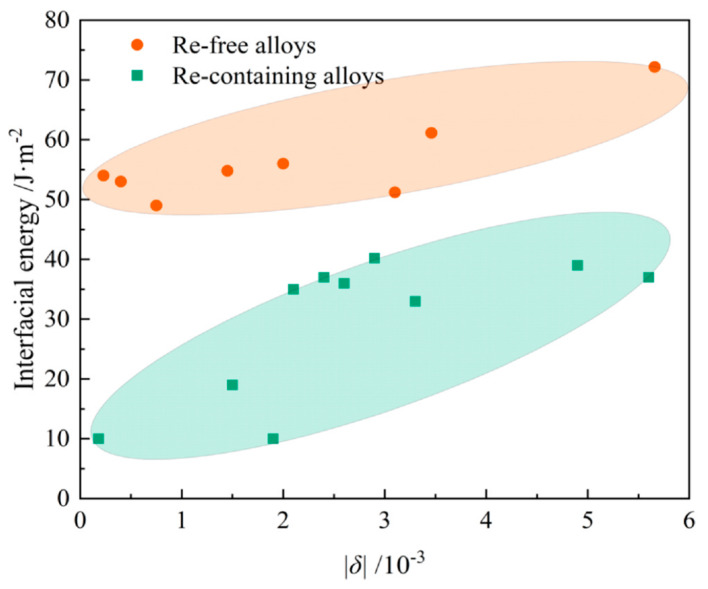
Lattice misfit dependence of interfacial energy in different alloys [57].

**Figure 13 materials-16-01787-f013:**
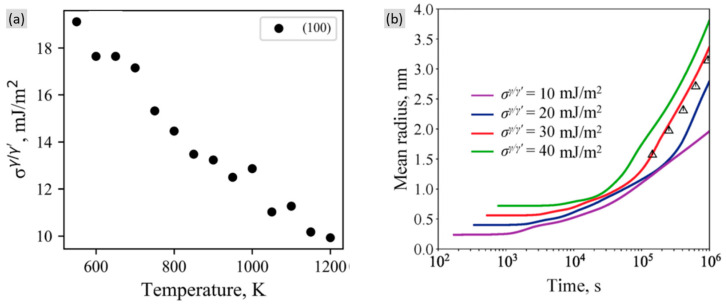
(**a**) Calculated interfacial energy varying with temperature and (**b**) calculated mean radius versus time at 773 K by using different interfacial energy of a Ni-Al binary alloy [59]. (The triangles show the experimental data in this reference.)

**Figure 14 materials-16-01787-f014:**
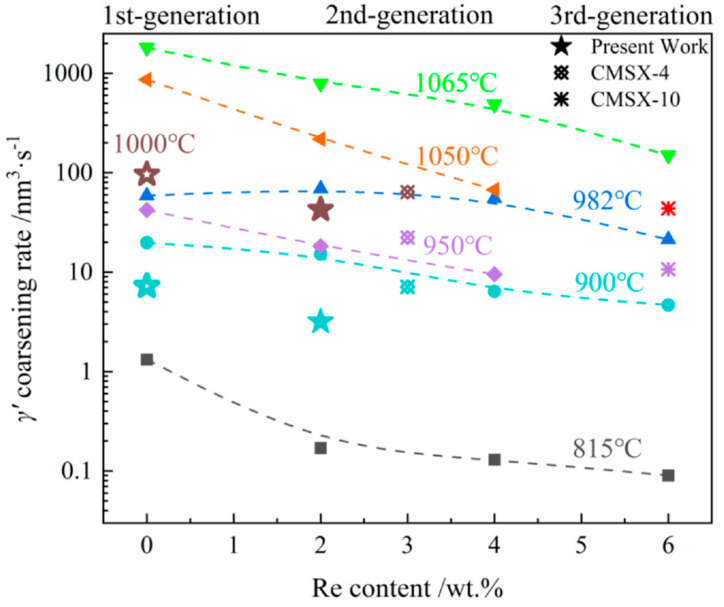
γ′ coarsening rate as a function of Re content at various temperatures [57]. (Different types of symbols indicated the data is extracted from different works.)

**Figure 15 materials-16-01787-f015:**
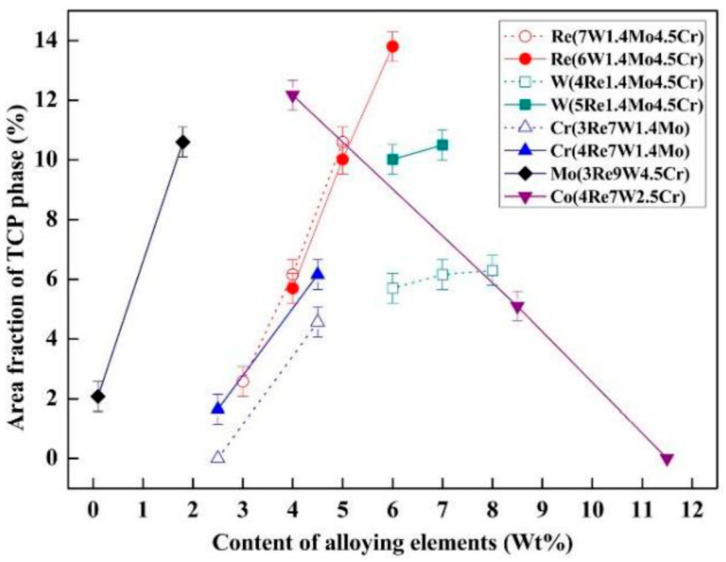
Area fraction of TCP phases of alloys with different alloying elements contents after thermal exposure at 1000 °C for 1000 h [60].

**Figure 16 materials-16-01787-f016:**
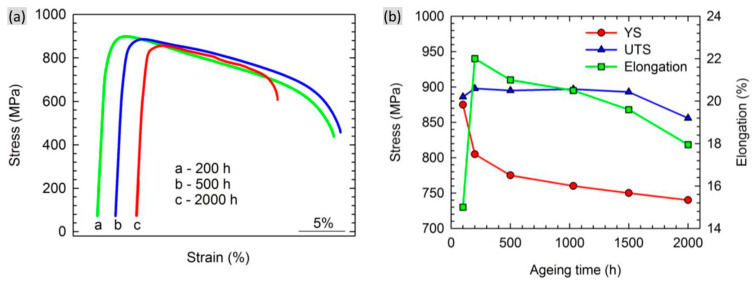
(**a**) Typical strain–stress curves at 950 °C of exposed samples for different time at 950 °C and (**b**) effect of thermal exposure time on 0.2% offset yield strength (YS), ultimate tensile strength (UTS) and plastic elongation to fracture of CMSX-4 alloy [61].

**Figure 17 materials-16-01787-f017:**
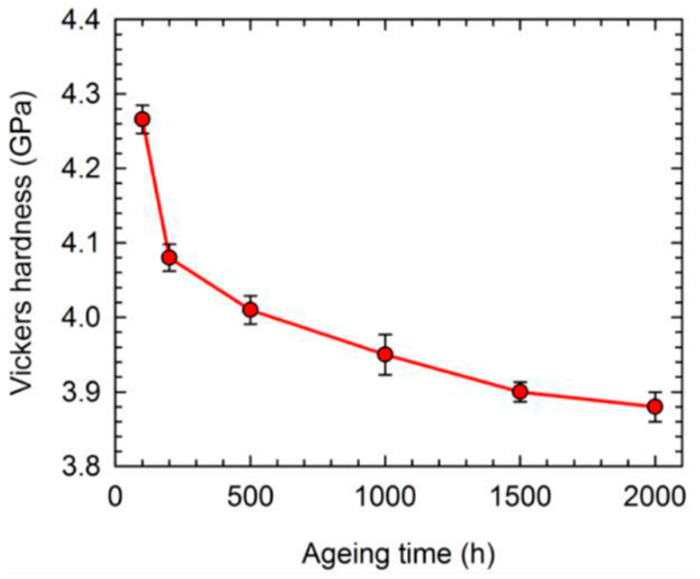
Variation in Vickers hardness with the thermal exposure time [61].

**Figure 18 materials-16-01787-f018:**
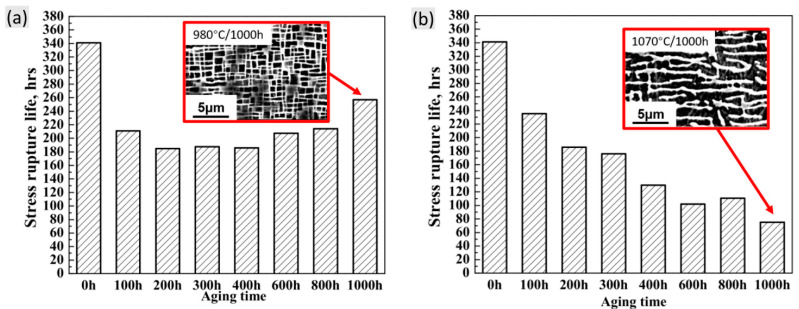
Effects of exposure time on the stress rupture life of DD6 alloy at 1070 °C/140 MPa after long-term exposure at (**a**) 980 °C and (**b**) 1070 °C. The associated microstructures after thermal exposure are also presented [34].

**Figure 19 materials-16-01787-f019:**
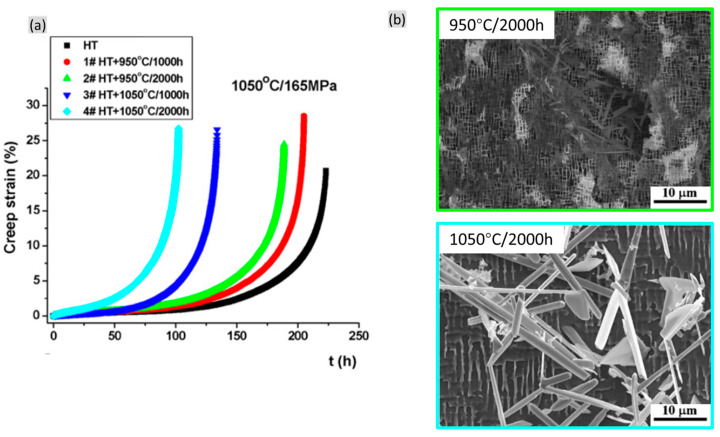
(**a**) Creep properties of specimens at HT state and after thermal exposure at different temperature and time of CMSX-4 alloy and (**b**) the associated microstructures after thermal exposure of 950 °C/1000 h and 1050 °C/1000 h [62].

**Figure 20 materials-16-01787-f020:**
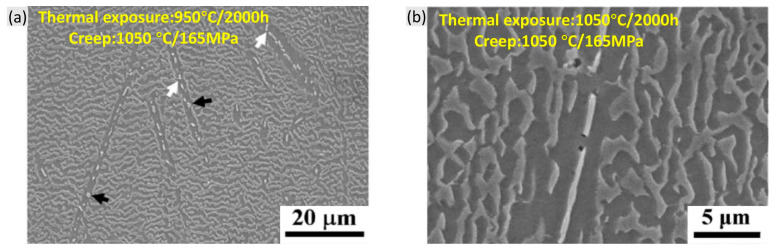
Microstructures in the crept specimens of CMSX-4 alloy after thermal exposure showing (**a**) rotated (white arrow) and coarsened (black arrow) μ particles free of crack and (**b**) crack initiation from the needle-like μ particle but without propagation [62].

**Figure 21 materials-16-01787-f021:**
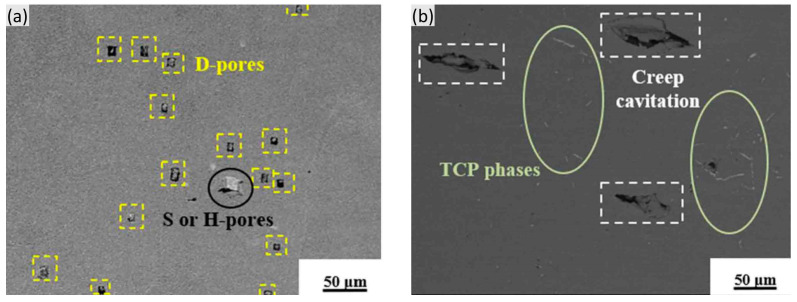
Microstructures in the crept specimen (at 1120 °C/137 MPa) of a Re-containing single crystal superalloy after thermal exposure of 1100 °C/500 h showing (**a**) the distribution of D-pores, and all of them marked by a yellow square dotted line, and (**b**) the crack distribution near the fracture surface, illustrating that the TCP phase does not cause crack directly [63].

**Figure 22 materials-16-01787-f022:**
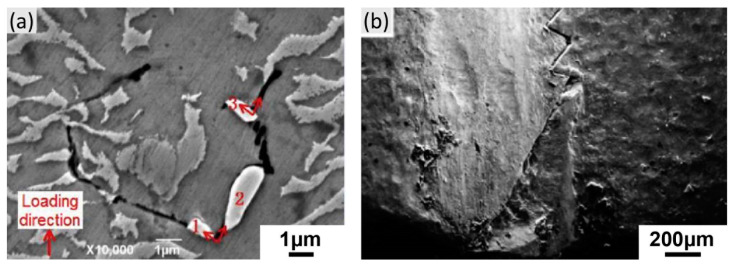
(**a**) The microcrack initiation and propagation near the TCP phase of DD6 alloy after creep at 1100 °C/140 MPa (Numbers 1,2 and 3 mark the TCP phases.) and (**b**) the fracture crack in the tenon of an aero-engine single crystal turbine blade [64].

## Data Availability

Not applicable.

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
