# Peer review of "An Overview of Thermal Exposure on Microstructural Degradation and Mechanical Properties in Ni-Based Single Crystal Superalloys"

_materials, 2023, doi:10.3390/ma16051787_

Round 1

Reviewer 1 Report

The present study represents an overview of thermal exposure on microstructural degradation and mechanical properties in Ni-based single crystal superalloys. Some revisions should be made in the study and they are summarized as follows:

- The paper looks well-organized but there are some grammatical mistakes in the paper. Therefore, the language of the paper should be checked.

- The importance of the study should be emphasized more intensively in introduction.

- In summary, it it is possiple, a future perspective for Ni-based single crystal superalloys can be drawn to show how their microstructural and mechanical properties will be increased.

Reviewer 3 Report

The paper reported An Overview of Thermal Exposure on Microstructural Degradation and Mechanical Properties in Ni-based Single Crystal 3 Superalloys. However, as a review paper, the work should be more detailed and up to date.

- The authors could explain in more detail the influence of precipitates that can change the mechanical behavior, not only as damage but as improvements. Does this happen at high temperatures and times reported?

- What does the literature report on treatment and service temperatures?

- How do superalloys stand out for these characteristics?

Round 2

Reviewer 2 Report

The paper in this form can be accepted.

Reviewer 3 Report

The authors have added the requested content. However, I believe that all reviewed responses could improve the study.